# Irreversible Electroporation for Prostate Cancer

**DOI:** 10.3390/life11060490

**Published:** 2021-05-27

**Authors:** Sean Ong, Matthew Leonardo, Thilakavathi Chengodu, Dominic Bagguley, Nathan Lawrentschuk

**Affiliations:** 1EJ Whitten Foundation Prostate Cancer Research Centre, Epworth HealthCare, Richmond, VIC 3121, Australia; sean.ong@student.unimelb.edu.au (S.O.); Thili.Chengodu@epworth.org.au (T.C.); dominic.bagguley@student.unimelb.edu.au (D.B.); 2Department of Surgery, University of Melbourne, Austin Health, Heidelberg, VIC 3084, Australia; matthew.leonardo@student.unimelb.edu.au; 3Fakultas Kedokteran, Universitas Indonesia, Jakarta Pusat 10430, Indonesia; 4Department of Urology, Royal Melbourne Hospital, Parkville, VIC 3050, Australia

**Keywords:** prostate cancer, irreversible electroporation, focal therapy, surgery

## Abstract

Although it can be lethal in its advanced stage, prostate cancer can be effectively treated when it is localised. Traditionally, radical prostatectomy (RP) or radiotherapy (RT) were used to treat all men with localised prostate cancer; however, this has significant risks of post-treatment side effects. Focal therapy has emerged as a potential form of treatment that can achieve similar oncological outcomes to radical treatment while preserving functional outcomes and decreasing rates of adverse effects. Irreversible electroporation (IRE) is one such form of focal therapy which utilises pulsatile electrical currents to ablate tissue. This modality of treatment is still in an early research phase, with studies showing that IRE is a safe procedure that can offer good short-term oncological outcomes whilst carrying a lower risk of poor functional outcomes. We believe that based on these results, future well-designed clinical trials are warranted to truly assess its efficacy in treating men with localised prostate cancer.

## 1. Introduction

Prostate cancer continues to be one of the most commonly diagnosed cancers in men and a leading cause of cancer deaths in males worldwide [1]. Although it can be lethal in its advanced stage, prostate cancer can be effectively treated when it is localised. Traditionally, radical prostatectomy (RP) or radiotherapy (RT) were used to treat all men with localised prostate cancer regardless of their risk. However, notwithstanding their improved overall survival benefits, RP and RT have significant risks of post-treatment side effects; the two most common being urinary incontinence (UI) and erectile dysfunction (ED).

The last decade has seen a change in management for very low and low risk prostate cancer and an increasing interest in new techniques for the treatment of intermediate risk prostate cancer. Very low and low risk prostate cancer has an unlikely chance of metastasis and a very low risk of mortality [1]. As such, guidelines now recommend treatment with active surveillance to delay or mitigate the need for RP or RT [2]. Similarly, some evidence suggests that intermediate risk prostate cancer can have oncological outcomes close to that of low risk prostate cancer [3]. However, the risk of metastasis is still relevant and warrants some form of treatment. Different forms of focal therapy to the prostate gland are currently being trialled for these men.

By focusing treatment on a specific area of the prostate gland, the aim of focal therapy is to achieve similar oncological outcomes to radical treatment while preserving functional outcomes and decreasing the rates of adverse effects. Irreversible electroporation (IRE) is a novel focal therapy modality which utilises pulsatile electrical currents to ablate tissue. Animal and human models have been used to prove that IRE can induce cell death whilst preserving important surrounding structures [4].

The mechanism in which IRE does this is through the destabilisation of the cell membrane, causing the alteration of membrane shape and the formation of nanopores. The excessive permeability of these cells disrupts the osmotic balance, leading to irreversible damage and the process of apoptosis [5]. This technique has now been refined to administer electrical pulses at levels to prompt cell death whilst keeping the procedure below harmful thermal thresholds [4]. For important structures such as blood vessels, IRE has been shown to decrease smooth muscle cells but maintain the connective tissue matrix [6]. Thus, it has been used effectively in liver lesions where damage to bile ducts and hepatic vessels are lethal [7,8]. For men with prostate cancer, preservation of the neurovascular bundles adjacent to the gland can result in preservation of continence and erectile function, therefore increasing quality of life.

Initial trials for localised prostate cancer patients show promising results in both oncological and functional outcomes, but more information on its clinical performance is required before clinicians can integrate IRE into routine clinical practice. This narrative review describes the IRE procedure, summarises the available data about irreversible electroporation as a focal therapy for prostate cancer, and discusses future perspectives in this field.

## 2. Methods

Google Scholar, Medline, and EMBASE databases were searched to identify evidence suited to the topic. The key words “irreversible electroporation”, “electroporation”, “NanoKnife”, and “prostate cancer” were used to standardise the search amongst the search engines. MeSH terms “Electroporation” and “Prostatic Neoplasms” were also included in the search. Only references published from 2011 to 2021 were included in this review. Relevant references were also identified from the studies and reviews found. A manual search for meeting abstracts and ongoing trials on clinicaltrials.gov was performed. The flowchart below summarises the included and excluded studies in this review (see Figure 1).

## 3. IRE Technique

The IRE procedure consists of different phases, including patient preparation, field visualisation, device set-up, needle insertion, and treatment delivery. All phases of the procedure have been described in the literature and are performed in an operating theatre [9,10,11,12].

Patients are transferred to the operating bed and administered general anaesthesia. The patient is then placed in a lithotomy position and a urethral catheter is inserted [12]. Following patient positioning, the next phase of the procedure is the visualisation of the treatment field. A trans-rectal ultrasound probe is inserted and the prostate’s three-dimensional measures are observed (axial, sagittal, and coronal planes) [9]. Anatomical landmarks, including the urethra, urinary bladder neck, prostatic–seminal vesicle angles, external sphincter, and the edge of the prostate base, are identified and marked in both the sagittal and axial plane [9,10]. The target lesion and its location relative to the other structures is also visualised in an axial plane grid [9].

Insertion of 19-gauge monopolar IRE electrode needles can then begin following the visualisation of the target lesion. Guided by imaging, the first needle is inserted, minding its parallel orientation and the distances of the apex and the base [9]. The needle is then guided in an axial view to ascertain its three-dimensional and relative position from structures such as the urethra and prostate capsule [9]. The remaining needles are inserted in the same way [9].

Of note, electrical pulses fired during an IRE procedure can cause unwanted muscle contractions; thus, the induction of deep muscle paralysis is vital for the precision of the procedure [9,10]. A muscle relaxant is administered prior to the connection to IRE device in preparation for the electrical pulses [9].

After insertion of the needles, the needles are connected to the IRE device. The device is a low energy direct current generator controlled by computer-based treatment planning software. The distance between electrodes is entered into the device software manually and it will provide a visualisation of the electrode placement and the estimated ablation zone in an axial image [10]. The device is then set to deliver 90 pulses in sets of 10, with a brief recharge time between each set. The duration of each pulse is 70 ms, separated by 100 ms, to achieve a current between 20–40 A between the electrode pairs [9]. Voltage is chosen depending on the distance between the electrodes; however, it is usually within 1200–1800 V/cm, with a maximum of 3000 V/cm. This range is optimal to ensure complete ablation without damage from heat, while also preventing undertreatment [9].

The initial 10 pulses serve as a “pulse test” to check for full muscle paralysis, as unwanted muscle contractions can alter the position of the needles and change the ablation zone [9]. The pulse test also verifies that the correct voltage is applied between the needles and ensures that the treatment will not reach the thermal threshold. Adjustments are made as necessary, and when optimised, the remaining 80 pulses are delivered [9].

When the needles are in place, the operator starts the device after entering the key parameters such as needle positions, relative needle distances, and active length exposure into the device [9]. The device automatically derives a voltage between the needles to produce an electric field. Throughout the treatment, the operator must hold the needles firmly to prevent ablation outside of the estimated zone [9]. Furthermore, keeping watch on both the sagittal and axial view is important to confirm that the ablated area is the intended one [9]. After the treatment, the needles are removed and the perineum is manually compressed for about 1 min [9].

## 4. Functional Outcomes of Irreversible Electroporation

The main goal for focal therapy treatment such as IRE is to provide good oncological control whilst preserving the patient’s quality of life. The two main functional side effects of radical prostate cancer treatment are erectile dysfunction (ED) and urinary incontinence (UI). Therefore, UI and ED are the most commonly used parameters in assessing functional outcomes in trials of focal therapy.

Currently, the literature demonstrates that IRE can preserve urinary and erectile function post-procedure at relatively high rates (Table 1).

Two retrospective analyses assessed the outcomes of prostate cancer patients treated with IRE. Valerio et al. analysed 34 men with localised prostate cancer. The median age of the men was 65.5 years. Outcomes were physician evaluated through patient reports of erections sufficient for intercourse and pad use. It was found that all patients who were continent pre-treatment had preserved continence post-treatment (24/24 men). Out of 20 men who had erectile function pre-treatment, 19 (95%) men had preserved erectile function at the median follow-up of 6 months [13]. Murray et al. explored functional outcomes in 25 low to intermediate risk prostate cancer patients. A standardised and validated questionnaire called the Prostate Quality of Life Survey was used to assess functional outcomes. The study found that 12/13 (92%) of patients retained erectile function, while 15/17 (88%) were pad-free during 12 months of follow-up [14].

Ting et al. conducted a prospective phase I–II trial in 2016 including 25 patients. Functional outcomes were measured using the Expanded Prostate Cancer Index Composite (EPIC) questionnaire. The study found that all participants (18/18) who completed the questionnaire had preserved erectile function and had pad and leak free continence at 6 months follow-up [15].

A 2017 phase IIa prospective study was conducted by Valerio et al. on 20 prostate cancer patients. This study utilised I-PSS (International Prostate Symptom Score), I-PSS QoL, UCLA-EPIC, EQ-5DTM, IIEF-15, FACT-P, and MAX-PC questionnaires. At 12 months follow-up, it was found that 100% of men (16/16) were pad-free and 83% (10/12) of initially potent patients retained erections sufficient for intercourse [16].

Also in 2017, Scheltema et al. prospectively analysed 18 patients who had undergone salvage therapy with IRE. The EPIC questionnaire, the American Urological Association (AUA) symptom score, and the SF-12 were used to assess patients’ quality of life. All patients that were pad-free at baseline remained continent at 6 months (8/8), while 50% (2/4) of patients retained erectile function at 12 months follow-up [17].

A phase II analysis of 63 patients was done by Van den Bos et al. in 2018. This study also used the EPIC questionnaire to assess quality of life and functional outcomes. Overall, all patients had pad-free continence, while 77% had preserved their potency at 12 months [18].

Collettini et al. carried out a prospective phase II study in 2019 on 30 biopsy proven low to intermediate risk prostate cancer patients. This study assessed functional outcomes using the IIEF-5 and a modified ICIQ-MLUTS questionnaire. The results showed that 96.5% of patients (28/29) were pad-free continent, while 79.3% of patients (23/29) retained erectile function at 12 months follow-up [19].

Blazevski et al. conducted two studies in 2020. The first was a biopsy-monitored prospective cohort study of 123 prostate cancer patients. This study used the EPIC questionnaire and AUA symptom score to assess quality of life and functional outcomes. The results found that 98.8% (80/81) of patients were pad-free, while 93% (49/53) of patients had erections firm enough for at least some sexual activity at 12 months follow-up [20].

The second study focused on patients with apical prostate lesions. Data on 50 patients from a prospective database were obtained and the EPIC questionnaire was used to measure functional outcomes. The results showed that 98% (49/50) of patients were pad-free, while 94% (30/32) remained potent at 12 months follow-up [21].

Overall, the studies mentioned above show that potency rates ranged from 50–100%. If the study by Scheltema et al. was removed, given that their study only included four men analysed for erectile function, then potency rates ranged from 77–100% at 6–12 months follow-up. This is an improved rate compared to studies of post-prostatectomy erectile function, which report potency rates of 29–80% and 44–93% at 3–24 months for unilateral and bilateral nerve sparing prostatectomy, respectively [22].

Regarding urinary continence, the above studies found a high rate of continence, ranging from 88–100% pad free continence rates from 6–12 months post-op follow-up. This compares well to urinary incontinence rates post-robotic assisted radical prostatectomy, which are reported to be around 4–31% [23].

## 5. Oncological Outcomes

Oncological outcomes for men post-IRE are based on a combination of pathology, imaging, and histology to identify the presence or absence of prostate cancer. A recommendation of focal therapy and salvage ablative therapy follow-up modalities was published based on an international Delphi consensus and an FDA consensus [24]. Their recommendations consist of serum prostate-specific antigen (PSA), multi-parametric magnetic resonance imaging (mpMRI), systematic and targeted prostate biopsies, as well as functional outcome questionnaires at various time points. Below, we summarise the oncological outcomes published so far in the literature (Table 2). Of note, a large study of 471 men [25] receiving IRE was not included here because their cohort included many men with high grade and nodal disease.

Colletini et al., analysed 30 men who received IRE for low to intermediate risk prostate cancer (National Comprehensive Cancer Network (NCCN) criteria). The median age of their cohort was 65.5 years and median PSA was 8.65 ng/mL (IQR 5–11 ng/mL). Most men (26/30) had intermediate risk prostate cancer. Post-IRE, median PSA levels decreased from 8.65 ng/mL to 2.35 ng/mL (IQR 1.1–3.4 ng/mL) at 12 months. Of the 28 men who received a 6-month prostate biopsy, two (7.14%) men had clinically significant (Gleason >3 + 3, core length > 3 mm) in-field recurrence and no men had out of field recurrence. Four men went on to have radical prostatectomy and one man had repeat IRE.

Similarly, Ting et al. analysed 25 men who had Gleason score < 8, lesion(s) visible on MRI, and no previous prostate cancer treatment. These men underwent the IRE procedure and were followed up with PSA and MRI at 6 months and prostate biopsy at 7 months [15]. Two men had low risk (D’Amico risk stratification) and 23 had intermediate risk disease. The mean age of the cohort was 67, and median PSA was 6.0 ng/mL (IQR 4.3–8.6). Follow-up at 6 months revealed a median PSA drop to 2.2 ng/mL. The 7-month biopsy showed a 0% (21/21) rate of infield recurrence; 19% (4/21) of men had recurrence adjacent to the ablated zone and 5% (1/21) of men had out of field recurrence [15]. Of the five men who had significant disease on biopsy, three remained on active surveillance, one underwent repeat IRE, and one progressed to robotic assisted radical prostatectomy [15].

In 2018, Van den Bos et al. published updated results from the database above. This time, 63 men were included in the analysis. Men received IRE and were followed up with PSA and MRI at 6 months and prostate biopsy at 6–12 months. The median age and PSA for this group was 67 and 6.0 ng/mL, respectively. Eight men had low risk disease and 55 men had intermediate risk disease (D’Amico risk classification). At the time of publication, 45/63 men had undergone post-IRE prostate biopsy. Seven men (15.6%) had significant in-field recurrence and four men (8.9%) had out-field recurrence. Of these men, four men began active surveillance, four men had repeat IRE, one had robotic assisted radical prostatectomy, and two underwent radiation therapy [18].

The most recent update of this database was published by Blazevski et al. in 2020. In total, 123 men were included in this analysis. Men received IRE and again were followed up with PSA and MRI at 6 months and prostate biopsy at 12 months. The median age of all men was 68 years, and median PSA was 5.725 ng/mL (3.8–8.0). It was found that 112 men (91%) had intermediate risk disease and 11 men had low risk disease (D’Amico risk classification). Median PSA fell from 5.725 ng/mL to 3.48 ng/mL (IQR 1.43–5.67) post-IRE. Of the 102 men who had undergone biopsy at the time of publication, 10 (9.8%) men had significant in-field recurrence and 13 (12.7%) had out of field recurrence. A total of 79 (77.5%) men had no significant disease found at their 12-month biopsy. Furthermore, analysis excluding the initial 32 patients (change in technique and improved skill) found an in-field recurrence rate of 2.7% (2/74 men) and out of field recurrence rate of 12.1% (9/74). The authors report that of these 74 men, two had repeat IRE and six went on to have salvage whole gland treatment [20].

Additionally, a subgroup analysis of men receiving IRE specifically for apical prostate tumours was published; 50 men were included in the analysis. Median PSA decreased from 6.25 ng/mL to 1.7 ng/mL. Of the 40 men who had a 12-month biopsy at this time, only one (2.5%) patient had in-field recurrence and eight (20%) men had out of field recurrence. Overall, 31 (78%) men were free of significant disease at their 12-month biopsy [21].

Scheltema et al. were the first to study IRE in the context of men with recurrent prostate cancer after radiotherapy treatment. A total of 18 men with localised radio-recurrent prostate cancer were recruited. The median age of these men was 71 years. Three men experienced biochemical failure (Phoenix definition) post-procedure. Of the 10 men who had a 12-month prostate biopsy, one (10%) had significant in-field recurrence and one (10%) had significant out of field recurrence [17].

## 6. Imaging Outcomes

The last decade has seen a surge in the use of multiparametric magnetic resonance imaging (MRI) of the prostate, culminating in its inclusion in multiple prostate cancer guidelines [2,26]. Whilst the main use of MRI is in the diagnostic pathway for newly diagnosed cancer, some evidence has shown that MRI is a useful tool for assessing the local recurrence of prostate cancer post-radical treatment as well [27,28]. Its strengths lie in the differentiation of post-treatment fibrosis and cancer, therefore enabling it to localise disease and aid with targeted prostate biopsies [29]. For IRE, studies have shown that treatment ablation can easily be seen on early post-treatment MRI [30,31]. However, its role in post-treatment follow-up and detection of cancer recurrence is still being investigated.

Scheltema et al., studied this particular topic [32]. This group analysed 50 men who underwent IRE and were followed up with prostate MRI at 6 months post-treatment. The accuracy of MRI was compared to prostate biopsy at 12 months. A total of 18 regions of interest in 17 patients were identified. Out of 33 patients who had a negative MRI, 10 men had cancer detected on prostate biopsy. The positive predictive value and negative predictive value of significant prostate cancer (Gleason score >3 + 4 or 4 mm) for in-field recurrence were 33% and 88%, respectively. For out of field recurrence, they were 50% and 80%, respectively, and for the whole gland, they were 47% and 70%, respectively [32]. These results suggest that MRI may be able to rule out significant disease; however, given the study’s small numbers, biopsy should still be performed to confirm the presence of prostate cancer until these results can be confirmed with a larger trial.

Contrast enhanced ultrasound (CEUS) is another imaging modality that has been analysed for post-IRE follow-up. This involves a trans-abdominal ultrasound performed after an intravenous bolus of 2 mL sulphur hexafluoride microbubbles. A retrospective analysis of 50 men receiving both CEUS and MRI showed that the sensitivity, specificity, positive predictive value, and negative predictive value of CEUS compared to MRI was 76%, 81%, 73%, and 83%, respectively [33].

## 7. Post-IRE Complications

Irreversible electroporation is a minimally invasive procedure, but post-treatment adverse outcomes can still occur in patients. The complications of IRE across several studies are summarised in Table 1. Overall, there were no severe adverse effects experienced by patients post-IRE. One exception would be a patient in the study by Ting et al. who experienced non-ST-elevated myocardial infarction after the procedure, but this is unrelated to the IRE procedure itself [15]. There were also a few cases of urethral stricture that required surgery [16,19].

Generally, patients experience grade I to grade II adverse effects according to the Clavien–Dindo classification and CTCAE. These adverse effects include mild haematuria, dysuria, urinary tract infections, pain, urgency, and temporary incontinence.

## 8. Conclusions and Future Directions

So far, early studies have shown that IRE is a safe procedure that can offer good short-term oncological outcomes whilst carrying a lower risk of poor functional outcomes compared to radical treatment. Based on these results, larger comparative phase three trials are warranted to further investigate the effects of IRE and to provide meaningful long-term data for these men. Ultimately, its effects on prostate cancer metastasis and survival will determine if IRE will become an option in clinical practice guidelines.

Two challenges that remain in trial design for focal therapy are the selection criteria for men that would most benefit from this treatment and the appropriate follow-up protocols for these men. Although the niche for focal therapy lies in the treatment of localised intermediate risk prostate cancer (given low risk disease can be treated with active surveillance and high risk disease has high recurrence rates, even with radical treatment), the delineation between low, intermediate and high risk prostate cancer is still grey, given the complexity of prostate cancer risk stratification. For IRE, the size of the lesion and the number of lesions within the prostate may be significant considerations when selecting patients due to its focal nature and the in-field and out of field recurrences reported in early studies of IRE. Higher risk histological features may also contribute to these recurrences. Furthermore, the efficacy of IRE for radio-recurrent prostate cancer needs to be established. Answers to these questions require more data, which will be facilitated by the development of registry databases such as The Clinical Research Office of the Endourological Society (CROES ClinicalTrials.gov identifier: NCT02255890) and Australasian IRE databases [5].

Another area of research that may impact the selection criteria and follow-up for men undergoing focal therapy treatment is the accurate identification of prostate cancer lesions. The accuracy of prostate MRI for the detection of cancer post-IRE still needs to be investigated. The addition of prostate specific membrane antigen positron emission tomography (PSMA PET) imaging may also improve the localisation of prostate cancer lesions and therefore help to characterise the prostate more accurately. Already proven in the staging of prostate cancer [34] and included in guidelines for biochemical recurrence [2], PSMA PET is now being investigated at the stage of initial diagnosis [35]. For IRE, PSMA PET has the potential to be used at the diagnosis stage to identify lesions missed by MRI, thereby helping to plan IRE probe placement. Furthermore, PSMA PET could be utilised in the follow-up of men post-IRE where recurrent lesions seen on PSMA PET imaging can be targeted by prostate biopsy.

To our knowledge, there is only one trial that is looking to compare IRE to radical prostatectomy (ClinicalTrials.gov identifier: NCT04278261). This group will recruit 438 men with localised prostate cancer (PSA < 20 ng/mL, T1a–T2c, Gleason score < 8) at a single centre in Shanghai. Their primary outcome measure will be 5-year tumour progression rate as seen by prostate biopsy, MRI and PSMA PET. Another group will look to compare focal IRE with extended (half-gland) IRE in 106 men with low to intermediate risk prostate cancer (ClinicalTrials.gov identifier: NCT01835977). Their primary outcome will be patient side effects and quality of life post-treatment. The results of these studies are eagerly awaited as clinicians and researchers look to improve the treatment paradigm for localised prostate cancer.

## Figures and Tables

**Figure 1 life-11-00490-f001:**
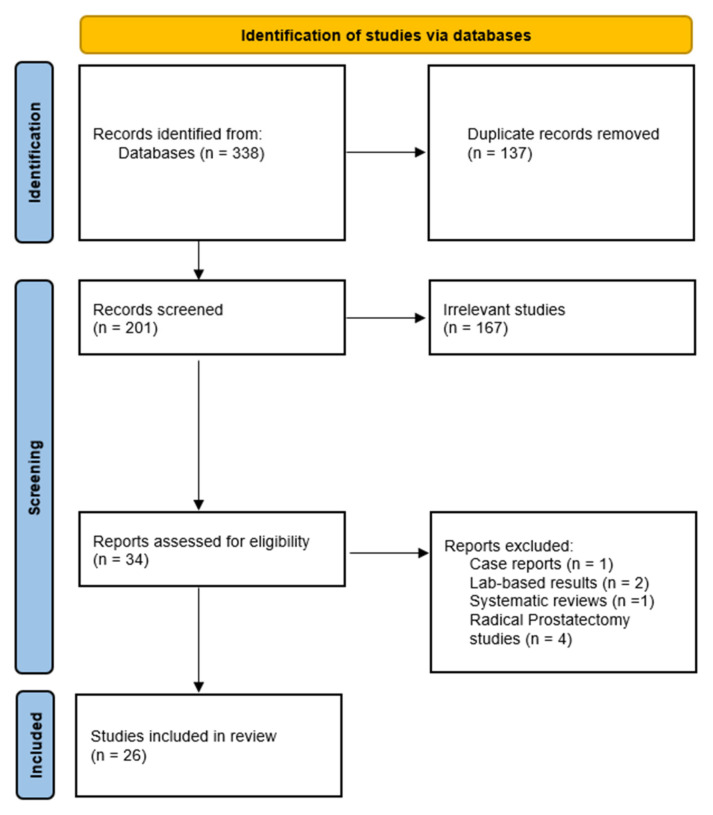
PRISMA flowchart of study selection process.

**Table 1 life-11-00490-t001:** Functional outcomes and complications.

Authors, Year	N=	Functional Outcomes		Complications
Valerio et al., 2014 [13]	34	Pre-IRE -	6 months: 24/24 continent (100%) 19/20 potent (95%)	2 urinary retention (6%) 6 debris/haematuria (18%) 5 dysuria (15%) 5 UTI (15%)
Murray et al., 2016 [14]	25	Pre-IRE 77% had urinary function score > 17 59% had erectile function score > 22	12 months 15/17 continent (88%) 12/13 potent (92%)	Clavien–Dindo Classification 6/27 grade 1 (22%) 7/27 grade 2 (26%) 1/27 grade 3 (4%)
Ting et al., 2016 [15]	25	Pre-IRE AUA urinary symptom score: 8 Sexual function score (EPIC): 56	6 months 0% change from baseline for continence and potency	Clavien–Dindo Classification 5 urinary retention (grade 1) (20%) 6 intermittent haematuria (24%) 1 nSTEMI (grade 3)
Valerio et al., 2016 [16]	20	Pre-IRE -	12 months 16/16 continent (100%) 10/12 potent (83%)	5 haematuria, dysuria (26.3%) 4 UTI (21%) 1 urethral stricture (5.2%)
Scheltema et al., 2017 [17]	50	Pre-IRE -	12 months 8/8 continent (100%) 2/4 potent (50%)	CTCAE 5/18 grade 1 2/18 grade 2
Van den Bos et al., 2018 [18]	63	Pre-IRE -	12 months 45/45 continent (100%) 10/13 potent (77%)	CTCAE 15 haematuria, dysuria, urgency (grade 1) (24%) 7 UTI, incontinence (grade 2) (11%)
Collettini et al., 2019 [19]	30	Pre-IRE 29/30 continent (96.7%) 25/30 potent (83.3%)	12 months 28/29 continent (96.5%) 23/29 potent (79.3%)	CTCAE 2 intermittent haematuria, grade 1 (6.7%) 3 UTI, grade 2 (10%) 1 urethral stricture requiring surgery, grade 3 (2%)
Blazevski et al., 2020 [20]	123	Pre-IRE -	12 months 80/81 continent (98.8%) 49/53 potent (93%)	Clavien–Dindo Classification 27 grade 1 (22%) 11 grade 2 (9%)
Blazevski et al., 2020 [21]	50	Pre-IRE -	12 months 38/40 continent (95%) 30/32 potent (94%)	Clavien–Dindo Classification 10 grade 1 (20%) 9 grade 2 (18%)

**Table 2 life-11-00490-t002:** Short-term oncological outcomes.

Authors, Year	N =	Inclusion Criteria	Patient Demographics	Median Post-Op PSA	In Field/Out of Field Recurrence	Number of Men Who Progressed to Whole Gland Therapy
Blazevski, et al., 2020 [21]	50	Prostate cancer lesion within 3 mm of apical capsule, treated with IRE	Median PSA: 6.25 (4.35–8.9) Median age: 68 (63–71) Gleason Score •3 + 3: 5 patients •3 + 4: 37 patients •4 + 3: 6 patients •4 + 4: 2 patients D’Amico risk classification •Low: 5 patients •Intermediate: 43 patients •High: 2 patients	1.8 (0.84–3.35)	In-field recurrence: 1 patient Out-field recurrence: 8 patients	4
Blazevski, et al., 2020 [20]	123	Low (high volume > 4 mm) to intermediate risk PCa (D’Amico) Gleason score ≤ 7 (ISUP ≤ 3) Unilateral or midline anterior/posterior index tumour, allowing single targeted ablative therapy PSA ≤ 15 ng/mL Life expectancy ≥ 10 years No previous treatment for PCa No previous androgen suppression treatment for PCa Minimum 12 month follow-up Multiple lesions which can be encompassed in one treatment	Median PSA: 5.725 (3.8–8.0) Median age: 68 (62–73) Gleason Score •3 + 3: 12 patients •3 + 4: 88 patients •4 + 3: 23 patients D’Amico risk classification •Low: 11 patients •Intermediate: 112 patients	2.5 (1.43–5.675)	In-field recurrence: 10 patients Out-field recurrence: 13 patients	6
Collettini et al., 2019 [19]	30	Age > 18 years PSA level ≤ 15 Gleason score ≤ 3 + 4 Clinical stage ≤ T2c Transperineal template biopsy, an MRI–US fusion-guided biopsy, or a standard transrectal US-guided biopsy (≥10 cores) Localised prostate cancer with no signs of extracapsular extension or lymph node metastases No previous radiation therapy for prostate cancer, previous or concomitant androgen suppression therapy, or previous focal therapy of the prostate	Median PSA: 8.65 (5–11) Median age: 65.5 (60–68.8) Gleason Score •3 + 3: 7 patients •3 + 4: 23 patients NCCN risk group •Low: 4 patients •Intermediate: 26 patients	2.35 (1–3)	In-field recurrence: 5 patients Out-field recurrence: 2 patients	1
Scheltema et al., 2017 [17]	18	Patients with unifocal, localised radio-recurrent PCa after LDR/PDR/HDR brachytherapy or external beam radiation therapy Any Gleason score/ISUP grade 1–5 Any number of positive cores/core involvement No evidence of metastatic or nodal disease If applicable, stop androgen deprivation therapy before treatment	Median PSA: 3.5 (3.2–8.4) Median age: 71 (68–75) Gleason Score •3 + 3: 0 patients •3 + 4: 6 patients •4 + 3: 5 patients •4 + 4: 2 patients •≥4 + 5: 5 patients	0.39 (0.04–0.43)	In-field recurrence: 1 patient Out-field recurrence: 1 patient	0
Ting et al., 2016 [15]	25 -	Age ≥ 40 years Visible lesion on mpMRI with no evidence of ECE or SVI Stage ≤ T2c on mpMRI Transperineal, TRUS, or MRI-guided biopsies correlating with the visible lesion on mpMRI Gleason score ≤ 7 on biopsy Low–intermediate risk disease (D’Amico criteria)	Median PSA: 6.0 (4.3–8.6) Median age: 67 (60–71) Gleason Score •3 + 3: 2 patients •3 + 4: 15 patients •4 + 3: 8 patients D’Amico risk classification •Low: 2 patients •Intermediate: 23 patients	2.2 (1.0–5.0)	In-field recurrence: 0 patients Out-field recurrence: 5 patients	1
Van den Bos et al., 2018 [18]	63	Low to intermediate risk PCa Gleason score ≤ 7 (ISUP grade ≤ 3) Unilateral or single midline anterior/posterior index tumour, allowing single targeted ablative therapy Life expectancy ≥ 10 years No previous treatment for PCa No previous androgen suppression/hormone treatment for PCa	Median PSA: 6.0 (3.2–8.4) Median age: 67 (61–71) Gleason Score •3 + 3: 9 patients •3 + 4: 38 patients •4 + 3: 16 patients D’Amico risk classification •Low: 8 patients •Intermediate: 55 patients	1.8 (0.96–4.8)	In-field recurrence: 7 patients Out-field recurrence: 4 patients	Not reported
** Guenther E, et al., 2019 [25]	429	Men with prostate cancer who would potentially benefit from IRE treatment of their PCa and who refused all types of standard therapy were included, with the primary goal of significant tumour mass reduction, and complete local tumour ablation if achievable. Patients with all stages of disease were included.	Mean PSA: 10–250 Mean age: 64 Gleason Score •3 + 3: 82 patients •3 + 4/4 + 3: 225 patients •4 + 4: 68 patients •5 + 3/3 + 5: 3 patients D’Amico risk classification •N/A: 4 patients •Low: 25 patients •Intermediate: 88 patients •High: 312 patients		In-field recurrence: 12 patients Out-field recurrence: 16 patients	

** Not included in narrative review discussions due to inclusion of high-risk prostate cancer patients; PCa: prostate cancer; PSA: prostate-specific antigen; IRE: irreversible electroporation; MRI: magnetic resonance imaging; US: ultrasound; ISUP: International Society of Urological Pathology.

## Data Availability

Not applicable.

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
