# Peer review of "Irreversible Electroporation for Prostate Cancer"

_life, 2021, doi:10.3390/life11060490_

Round 1

Reviewer 1 Report

The authors belong to Prostate cancer research, Dept. of Surgery and Urology-they could and should provide more insights into this electrical technique-biologically, physiologically, and pathologically, maybe morphologically too.

Revise 2. Surgical Technique to 2, IRE Technique

Need pulse generator used and voltage or electric field applied in each case.

90 pulses were applied, at what interval? need interval between the busts of 10 pulses and interval between each pulse in the 10 pulse bursts.

Author Response

Thank you for your comments. We have responded as below

Comment 1: The authors belong to Prostate cancer research, Dept. of Surgery and Urology-they could and should provide more insights into this electrical technique-biologically, physiologically, and pathologically, maybe morphologically too.

We have now added the paragraph below to give some context.

“The mechanism in which IRE does this is through the destabilisation of the cell membrane causing the alteration of membrane shape and the formation of nanopores. Excessive permeability of these cells disrupts the osmotic balance leading to irreversible damage and the process of apoptosis.[Blazevski, 2020 #32] This technique has now been refined to administer electrical pulses at levels to prompt cell death whilst keeping the procedure below harmful thermal thresholds.[Davalos, 2005 #51] For important structures such as blood vessels, IRE has been shown to decrease smooth muscle cells but maintain connective tissue matrix.[Edd, 2006 #52]”

Comment 2: Revise 2. Surgical Technique to 2, IRE Technique

We have now changed this to IRE technique

Comment 3: Need pulse generator used and voltage or electric field applied in each case.

We have now mentioned “19-gauge monopolar electrode needles” and “low energy direct current generator controlled by computer based treatment planning software”.  

We have now added this in terms of voltage. “Voltage is chosen depending on distance between electrodes however it is usually within 1200-1800V/cm with a maximum of 3000V/cm.”

Comment 4: 90 pulses were applied, at what interval? Need interval between the busts of 10 pulses and interval between each pulse in the 10 pulse bursts.

The paragraph now read like this.

“The device is then set to deliver 90 pulses in sets of 10 with a brief recharge time between each set. The duration of each pulse is 70ms each, separated by 100ms, to achieve a current between 20-40 A between electrode pairs.[7] Voltage is chosen depending on distance between electrodes however it is usually within 1200-1800V/cm with a maximum of 3000V/cm.”

Reviewer 2 Report

My review in the attached file.

Author Response

Thank you for your comments. We have responded as below.

Comment 1: In the introduction section rows 22-23 it’s need a reference.

We have added references for these sentences.

Comment 2: The rows 53-59 should be moved in a methods section that is absent.

We have added a new Methods section for this paragraph.

Comment 3: A methods section should be inserted including search criteria with the indication of temporal period, inclusion and exclusion criteria.

We have added the search criteria including keywords, MeSH term used and time period. It reads as this:

“Google scholar, Medline and EMBASE databases were searched to identify evidence suited to the topics. Key words “irreversible electroporation”, “electroporation”, “NanoKnife” and “prostate cancer” were used to standardise the search amongst the search engines. MeSH terms “Electroporation” and “Prostatic Neoplasms” were also included in the search. Only references published from 2011 to 2021 were included in this review. Relevant references were also identified from the studies and reviews found. A manual search for meeting abstracts and ongoing trials on clinicaltrials.gov was performed. The flowchart below summarized the included and excluded studies in this review (see Figure 1).”

Comment 4: A flow diagram should be inserted with the data about the analysed, included and excluded manuscripts

The PRISMA flow diagram below has been added.

Comment 5: The IRE procedure consists of different phases, starting with patient preparation, field visualization, device set-up, needle insertion and treatment delivery and has been described in the literature [7-10]. All of the phases of the procedure have been described in the literature and are performed in an operating theatre.

The concept is repeated. Please rephrase,

A possibility could be: The IRE procedure consists of different phases, starting with patient preparation, field visualization, device set-up, needle insertion and treatment delivery. All of the phases of the procedure have been described in the literature and are performed in an operating theatre [7-10].

We have used your rephrasing in our manuscript, thank you.

Comment 6: Please don’t start sentences with number.

All sentences that start with numbers have been changed.

Comment 7: Do not use abbreviations for procedures that appear for the first time in the text. Some abbreviations were not defined (e.g. mpMRI, PSA, etc).

Abbreviations were expanded the first time they appear in text.

Comment 8: Also the abbreviations list do not include all used abbreviations.

The abbreviation list below the table only accounts for the abbreviations used in the table. There is no dedicated abbreviation list in the manuscript template.

Comment 9: Authors et al must always be followed by the point.

We have made appropriate changes to this.

Comment 10: In the Table 1, I suggest to remove the title but to include the reference number.

We have removed the title column and replaced it with sample size instead. Reference numbers have also been added.

Reviewer 3 Report

Good summary of available data on IRE in prostate cancer.

My points: 

  1. Line 46: do you mean continence instead of "UI" here?
  2. Chapter Surgical Technique: i miss the description of different ablation zones i.e. focal vs. half gland vs. whole gland. How did the studies reported here differed on this aspect?
  3. Line 99: is instead of "are"
  4. Line 187: do you mean nadir instead of "drop"?
  5. Are there any data on IRE as a salvage tx after HIFU or for metastasis-directed tx?
  6. Please add the respective column with the number of treated pts. in Table 1

Author Response

Thank you for your comments. Below are our responses.

Comment 1: Line 46: do you mean continence instead of "UI" here?

Thank you, we have now changed this to “continence”

Comment 2: Chapter Surgical Technique: i miss the description of different ablation zones i.e. focal vs. half gland vs. whole gland. How did the studies reported here differed on this aspect?

All the studies mentioned here are for “focal therapy”.

Comment 3: Line 99: is instead of "are"

We have now changed this thank you

Comment 4: Line 187: do you mean nadir instead of "drop"?

We have mentioned drop here because the PSA may drop further on subsequent follow up which would be the nadir. The follow up was only at 6-12 months.

Comment 5: Are there any data on IRE as a salvage tx after HIFU or for metastasis-directed tx?

To our knowledge studies of IRE in post radiation men is the only salvage setting that has been published. No data on metastasis directed therapy using IRE has been published.

Comment 6: Please add the respective column with the number of treated pts. in Table 1

This has now been added.

Round 2

Reviewer 2 Report

No further comments

Reviewer 3 Report

My points have been sufficiently addressed.